# Effects of Dietary Supplementation of *Lacticaseibacillus Chiayiensis* AACE3 on Hepatic Antioxidant Capacity, Immune Factors and Gut Microbiology in Nandan Yao Chicks

**DOI:** 10.3390/antibiotics12091356

**Published:** 2023-08-23

**Authors:** Xin Kang, Xin-Dong Li, Cheng-Ying Luo, Wei-Gang Xin, Huan-Yu Zhou, Feng Wang, Lian-Bing Lin

**Affiliations:** 1Faculty of Life Science and Technology, Kunming University of Science and Technology, Kunming 650500, China; 2Engineering Research Center for Replacement Technology of Feed Antibiotics of Yunnan College, Kunming 650500, China

**Keywords:** *Lacticaseibacillus chiayiensis* AACE3, aureomycin, hepatic antioxidant capacity, immune factors, gut microbiology, chicks

## Abstract

The growing issue of antibiotic resistance has restrained the utilization of antibiotics as growth enhancers in the poultry industry. Probiotics are candidates for replacing antibiotics in the poultry industry. However, probiotics are strain-specific and their efficacy needs to be investigated before applying them. The aim of this study was to assess the positive effects of *Lacticaseibacillus chiayiensis* AACE3 on the health and gut microbiota of Nandan Yao chicks. The results showed that compared with the blank control (NC) and aureomycin (PC) groups, *L. chiayiensis* AACE3 increased final body weight (BW), villus height and improved the ratio of villus height to crypt depth in chicken jejunal tissues. *L. chiayiensis* AACE3 also increased the activity of hepatic antioxidant enzymes (SOD, CAT and T-AOC) and reduced hepatic oxidative damage (MDA). Furthermore, compared to NC, *L. chiayiensis* AACE3, the activity of intestinal digestive enzymes (i.e., α-amylase, lipase and trypsin) was increased. *L. chiayiensis* AACE3 upregulated the production of IgA and IgG and downregulated the production of IL-6, IL-1β and TNF-α in chicken serum. Moreover, supplementation of *L. chiayiensis* AACE3 enhances the diversity of gut microbes. At the phylum level, the abundance of Actinobacteriota and Proteobacteria decreased with *L. chiayiensis* AACE3 supplementation, while the abundance of Verrucomicrobiota and Bacteroidetes increased. At the genus level, there was an increase in the abundance of potential probiotics *Akkermansia*, *Romboutsia*, *Subdoligranulum*, and *Lactobacillus*. This study confirms that *L. chiayiensis* AACE3 is an excellent feed additive as an alternative to aureomycin and offers various advantages for the healthy growth of chickens during the brooding period by positively affecting their gut microbiome.

## 1. Introduction

In recent years, nutritional strategies have emerged as important for improving animal health [1,2]. Feed additives promote animal digestion and health [3]. Antibiotics are frequently utilized as feed supplements in poultry farming to manage and prevent diseases, as well as to promote the growth performance of animals [4]. However, there is a worldwide effort to reduce the use of antibiotics in poultry production, as antibiotic residues in poultry products can be harmful to consumers [5]. In addition, antibiotics can disrupt the balance of animal intestinal flora and affect the health of the animal’s intestinal tract [6]. The severe negative effects of antibiotics on livestock and poultry have led to their prohibition as feed additives in Europe, the United States, and China [7]. Today, there are new approaches being used to replace antibiotics in poultry farming. For example, vaccination can prevent proliferative enteropathy (PE) and reduce the consumption of oxytetracykline used to treat PE [8]. Mannanoligosaccharides (Bio-Mos®) can be used as an alternative to antibiotic growth promoters (avilamycin) in broiler feeding [9]. Sodium butyrate can also be used as an alternative to dietary hyoscine in chickens [10]. Moreover, probiotics are a promising alternative to antibiotics in broiler farming [11]. Probiotics are living microorganisms, and consuming adequate amounts of them can provide sustained health benefits for animals [12]. For instance, chickens that received *Lactobacillus plantarum* LP-8 dietary supplementation displayed increased intestinal mucosal immunoglobulin IgA content and an improved antioxidant capacity [13]. *Lactobacillus casei* supplementation resulted in significant enhancement of the immunity of chicks, modulation of the expression of inflammation-related factors, and increased gut microbiota diversity [14]. These studies have shown that probiotics can enhance host immunity while enriching the gut flora to promote host health. A large number of studies have proved that probiotics have good effects in animal breeding, but due to the different kinds of probiotic strains, their vitality and their adaptability to different environments, leads to the different effects of probiotics [15]. Therefore, probiotics that improve host health are essential for poultry farming.

*Lacticaseibacillus chiayiensis*, a novel LAB closely related to the *Lacticaseibacillus casei* group, was first isolated from cattle manure in Chiayi City (Taiwan, China) in 2018 [16]. Kim et al. found that the *L. chiayiensis* genome encodes stress resistance-related (e.g., bile, acid, and oxidative stress tolerance genes), bacteriocin production and adhesion-related genes. Furthermore, in an in vitro study of probiotic properties, *L. chiayiensis* showed good antibacterial properties, adhesion properties and an ability to survive in adverse gastrointestinal environments [17]. Therefore, *L. chiayiensis* can be considered a novel potential probiotic that may have good potential for application.

Resistance to pathogen colonization and local immune responses in the gut can be enhanced in neonatal birds by adding probiotics to the diet prior to the maturation of the gut microbiota [18,19]. Nandan Yao chicken is a breed of chicken native to China. It is characterized by coarse-grained tolerance, strong foraging ability, and delicate and tasty meat [20]. Nandan Yao chicken was selected into the National Livestock Genetic Resource Breeds List in 2021, and was awarded the National Geographical Indication Protection Certification for Agricultural Products in 2022, which has high economic value [21]. Nandan Yao chicken was used as a research material to evaluate the beneficial performance of *L. chiayiensis* in this study, which guided poultry breeding in China to a certain extent. It also provides a new possible option for antibiotic replacement in global poultry farming. Certainly, we will also select more modern broiler strains to evaluate the generalizability of the beneficial effects of *L. chiayiensis* in the future. In this study, we hypothesized that supplementation with *L. chiayiensis* could replace aureomycin to promote microbial proliferation and persistence to improve the immunity of chicks to improve the early development of their intestinal tract. Therefore, we evaluated the intestinal morphology, liver antioxidant function, immune factor content and intestinal microbiota of Nandan Yao chickens after *L. chiayiensis* supplementation. 

## 2. Results

### 2.1. Growth Performance

Table 1 shows the effect of the experimental group on body weight, average daily feed intake, average daily gain, and feed conversion ratio of the Nandan-Yao chickens. The chicks were weighed after 21 days, and the PC, LC6 and LC7 groups were found to be 11.28 g, 4.56 g and 14.69 g heavier than the NC group, respectively. ADFI and ADG for chicks were observed and found to be significantly higher in the PC and LC7 groups than in the NC group. However, the comparison of the FCR with the NC group does not yield a significant difference with respect to all experimental groups.

### 2.2. Effects of L. chiayiensis AACE3 on the Gut Morphology of Chickens

The use of *L. chiayiensis* AACE3 as a feed supplement significantly influenced the intestinal morphology of chickens (Figure 1). The villus height of the jejunum was significantly higher in the *L. chiayiensis* AACE3 group than in the NC and PC groups (*p* < 0.05). Regarding the VH:CD ratio, which represents the functional status of the epithelium, the LC7 group produced a significantly higher ratio than the control groups (*p* < 0.05), while it was increased by 5.17% relative to that with the PC group.

### 2.3. The Effect of L. chiayiensis AACE3 on Digestion Properties

Supplementation with *L. chiayiensis* AACE3 led to a significant increase in α-amylase and lipase activity relative to the NC (*p* < 0.05). The LC7 group showed a significant improvement in trypsin activity, while the LC6 group showed 58.79% accretion. Compared with the PC group, supplementation with *L. chiayiensis* AACE3 caused no significant difference in α-amylase, lipase, and trypsin activity (Figure 2).

### 2.4. Effects of L. chiayiensis AACE3 Supplementation on Hepatic Function

The effect of supplementation with *L. chiayiensis* AACE3 on the antioxidant capacity of chicken hepatic tissue is demonstrated in Figure 3. The levels of oxidative damage products (MDA) were significantly lower in chickens fed *L. chiayiensis* AACE3 supplementation compared to NC-treated chickens (*p* < 0.05). Compared to the PC group, MDA levels in the LC6 and LC7 groups decreased by 6.39% and 23.84%, respectively.

The liver SOD and GSH-Px activities were significantly increased in chickens fed *L. chiayiensis* AACE3 compared to the NC group. CAT and SOD activity was significantly increased in chickens fed *L. chiayiensis* AACE3 compared to the PC group. T-AOC activity was significantly increased in the LC7 group compared to the NC and PC groups by 22.86% and 17.27%, respectively. *L. chiayiensis* AACE3 effectively mitigates oxidative stress in chicken livers.

### 2.5. Effects of L. chiayiensis AACE3 on Immune Factors

Changes in the level of immune factors have an impact on chicken health to some extent, and supplementation with *L. chiayiensis* AACE3 induced the production of immune factors in chickens (Figure 4). Compared to the control (NC) group, supplementation with *L. chiayiensis* AACE3 significantly increased the concentration of IgA (*p* < 0.05). In addition, the LC6 and LC7 groups increased IgG concentrations by 13.76% and 10.79%, respectively. Compared to the aureomycin group (PC), the LC6 and LC7 groups increased IgA concentrations by 26.96% and 19.56%, respectively.

The LC7 group significantly reduced IL-6 concentrations relative to those with the NC and PC groups (*p* < 0.05). While IL-6 levels were reduced by 5.5% in the LC6 group, there was no significant change. In addition, the LC6 and LC7 groups reduced IL-1β and TNF-α levels by 14%, and 20.98% and 10.16%, and 14.98%, respectively, relative to those with the NC group. LC6 and LC7 groups reduced IL-1β and TNF-α levels by 9.5%, and 7.71% and 5.46%, and 0.71%, respectively, relative to those with the aureomycin group (PC). Thus, the results suggest that supplementation with *L. chiayiensis* AACE3 altered the levels of these immune factors.

### 2.6. Effects of L. chiayiensis AACE3 on the Chicken Gut Microbiota

Changes in the alpha-diversity index of gut microorganisms reflect changes in microbial community abundance and diversity (Figure 5). Significant differences were found between the community diversity (Simpson) indices of the NC and LC7 groups. Principal coordinate analysis (PCoA) and non-metric multidimensional scaling (NMDS) using the Bray distance metric demonstrated that the biological communities of the NC, PC, LC6, and LC7 groups were segregated into four distinct clusters (Figure 6). The reliability of the model was evaluated based on the stress value, which was 0.052 in NMDS. The findings suggest that the administration of *L. chiayiensis* AACE3 leads to several alterations in gut microbial composition.

At the phylum level, the most prevalent phyla are Firmicutes, Actinobacteriota, Verrucomicrobiota, Bacteroidetes, and Proteobacteria. Firmicutes was more abundant in the LC6 group, comprising 79.62% of the phylum-level community, while Actinobacteriota was more abundant in the NC and PC groups than in the experimental group. In the LC7 group, Verrucomicrobiota and Bacteroidota are more abundant than in the other groups and accounted for 22.08% and 13.03%, respectively, of the community at the phylum level. The abundance of Proteobacteria in the control group was higher than that in the remaining groups (Figure 7).

At the genus level, the top 10 dominant bacterial genera were identified (Figure 8). Among them, *unclassified Lachnospiraceae* was the most dominant bacterial genus, accounting for 13.25% of the community among the top 10 OTUs. *Unclassified Lachnospiraceae* levels were significantly higher in the LC6 group than in the PC group (*p* < 0.05) and in the LC7 group than in the NC group (*p* < 0.05). The levels of *Akkermansia* were significantly higher in the LC7 group (*p* < 0.05) than in the other groups. *Romboutsia* levels in the experimental groups (LC6 and LC7) were significantly higher (*p* < 0.05) compared to the NC and PC groups. The levels of *Subdoligranulum* were significantly higher in the LC6 group than in other groups (*p* < 0.05), while the level of *Collinsella* was significantly lower in the experimental LC6 and LC7 groups (*p* < 0.05) than in the NC group. Additionally, the levels of *Olsenella* and *Ruminococcus torques* were significantly higher in the PC group than in the other groups (*p* < 0.05). However, the levels of *Bifidobacterium*, *Blautia*, and *Lactobacillus* in the experimental group did not significantly differ from those in the control group. These results indicate that the relative abundance of some dominant bacteria in the intestinal flora changed after supplementation with *L. chiayiensis* AACE3. 

### 2.7. Correlation Analysis

To further explore the relationship between changes in the abundance of genera in the chicken gut microbiota and changes in the chicken’s own physiology. Correlation analysis of microorganisms (10 genera with the highest relative abundance) with digestive system characteristics, antioxidant capacity and immune status was performed (Figure 9). In the correlation analysis, *Ruminococcus torques* showed a significant negative correlation with T-AOC, SOD, CAT, and trypsin and a significant positive correlation with IL-6 and MDA. *Unclassified Lachnospiraceae* showed a significant negative correlation with GSH-Px and SOD and a significant positive correlation with IL-6 and MDA. *Collinsella* exhibited a considerable negative correlation with IgG and α-amylase and a significant positive correlation with TNF-α. Similarly, *Bifidobacterium* displayed a significant negative correlation with GSH-Px and α-amylase. *Olsenella* was significantly and positively linked with α-amylase. *Lactobacillus* showed a significant negative correlation with CAT. Moreover, *Romboutsia* showed a significant negative correlation with TNF-α and a significant positive correlation with IgA, lipase, trypsin, and SOD. On the other hand, *Akkermansia* showed a noteworthy negative correlation with IL-6 and MDA and a significant positive correlation with T-AOC, trypsin, and SOD. Finally, *Blautia* demonstrated a significant positive correlation with α-amylase, whereas no significant correlations were found with *Subdoligranulum*.

## 3. Discussion

In the field of poultry farming, probiotics are receiving increasing attention as an alternative to antibiotics [22,23]. The discovery of new probiotics and the evaluation of their role in poultry farming is essential to meet the demand for alternative antibiotics in poultry farming. In this study, the beneficial effects of *L. chiayiensis* on poultry farming were comprehensively evaluated through animal feeding experiments (addition of *L. chiayiensis* to diets) on Nandan Yao chickens. It provides a new option for antibiotic replacement in the field of poultry farming. In this study, the use of probiotic strain *L. chiayiensis* AACE3 significantly increased the body weight of chickens in the LC7 group. This aroused our interest in exploring how its growth-promoting ability is realized. Through the detection of digestive enzyme activities and intestinal morphology in the intestine, it was found that *L. chiayiensis* mediated the upregulation of intestinal digestive enzyme activities and the alteration of intestinal morphology in 21 d Nandan scallop chickens. The height of the intestinal villi of chickens fed *L. chiayiensis* was significantly increased compared to the control, while the depth of crypts was not significantly changed, whereas, in previous studies, it has been shown that the higher the ratio of villus height to crypt depth, the higher the nutrient digestibility [24,25]. In addition, another study suggested that the improvement in weight gain may be related to the ability of probiotics to secrete enzymes such as amylase, protease and lipase, which target the efficient absorption of feed nutrients such as starch, fat and protein [24]. In this study, *L. chiayiensis* AACE3 significantly increased α-amylase, lipase, and trypsin activities relative to the control group. Thus, these findings suggest that *L. chiayiensis* AACE3 enhances host digestion by promoting both host digestive enzyme production and morphological changes in intestinal villi. 

In addition, the antioxidant status within the host is essential to protect against pathogens and maintain homeostasis in vivo [25] as the body tends to produce reactive oxygen species (ROS) through normal cellular metabolism and in response to factors such as environmental changes and diet [26]. Additionally, the accumulation of excessive ROS can lead to host oxidative stress and oxidative damage. In this study, supplementation with *L. chiayiensis* AACE3 resulted in increased SOD, CAT, GSH-Px, and T-AOC levels and decreased MDA levels compared to the NC and PC groups. Among them, MDA is an important product of lipid peroxidation, and the level of MDA is a measure of the degree of oxidative damage [27]. Therefore, dietary supplementation with *L. chiayiensis* AACE3 significantly increased the antioxidant capacity of Nandan Yao chickens. Meanwhile, the significant reduction in MDA level also confirmed the repair of oxidative damage by the increase in antioxidant capacity induced by *L. chiayiensis*. Furthermore, it is important to note that the host’s ability to resist disease is more predominantly related to its own immune system [28]. Probiotics can prevent the proliferation of pathogens and thus prevent infection by improving physical mechanisms (immune organs) and chemical barriers (immune proteins, tiny molecules, etc.), contributing to host immune regulation [29]. For example, supplementation with probiotics results in higher levels of IgG and IgA, which are important markers for estimating changes in the immune function of the animal [30]. Here, supplementation with *L. chiayiensis* AACE3 increased the level of IgA and IgG levels in chicken serum. In addition, probiotics not only stimulate humoral and cell-mediated immunity (IgG and IgA) but also modulate the production of pro- and anti-inflammatory cytokines (IL-2 and IL-10) in the host, thereby improving host resistance to disease [31]. In this study, the levels of IL-1β, IL-6 and TNF-α were decreased by supplementation with *L. chiayiensis* AACE3. Overall, *L. chiayiensis* AACE3 improves poultry health by increasing the activity of antioxidant enzymes, stimulating humoral immunity, reducing MDA levels, and regulating the production of inflammatory factors. 

There is growing evidence that gut microbes positively impact host health [32]. In this study, 16S RNA from the cecal content of chickens fed *L. chiayiensis* AACE3 was sequenced to investigate changes in microbial community composition. The α-diversity analysis showed altered diversity after supplementation with *L. chiayiensis* AACE3, which may be because *L. chiayiensis* AACE3 inhibits harmful bacteria in the intestine [33]. PCoA and NMDS showed different β diversity between supplementation with *L. chiayiensis* AACE3 chickens and the NC and PC groups, indicating that the effect of supplementation with *L. chiayiensis* AACE3 on the intestinal flora was different from that of the NC and PC groups. In addition, microorganisms play different roles in various hosts. Therefore, it is also important to analyze the changes in the high abundance of microorganisms in the gut microbiota that have a greater impact on the host [34]. In a phylum-level analysis, we found that the abundance of the phyla Actinobacteriota and Proteobacteria declined after feeding with *L. chiayiensis* AACE3. Proteobacteria consist of various zoonotic pathogens, including but not limited to *Escherichia*, *Salmonella*, and *Campylobacter*, as well as other highly virulent genera [35]. This study observed an improvement in Verrucomicrobiota and Bacteroidota abundance upon supplementing *L. chiayiensis* AACE3, which are important in energy production and metabolic processes within the gut [36]. *Akkermansia* is a representative species of Verrucomicrobiota, an anaerobic bacterium that degrades intestinal mucus, preserving the integrity of the intestine and avoiding the development of necrotizing small bowel inflammation (intestinal necrosis) [37,38]. In addition, the addition of *L. chiayiensis* AACE3 to the gut microbiota altered the composition of dominant bacterial species at the genus level. Among them, the levels of the dominant genera *Akkermansia*, *Romboutsia* and *Subdoligranulum* were increased significantly, and the levels of *unclassified Lachnospiraceae*, *Ruminococcus torques group* and *Collinsella* decreased significantly. Based on a correlation analysis, *Akkermansia* relative abundance was found to be positively correlated with trypsin content and T-AOC while negatively correlated with IL-6 levels in chickens. It is similar to some of the other studies on *Akkermansia*, namely that changes in *Akkermansia* levels affect host metabolic processes and reduce inflammatory responses [39]. *Romboutsia* was positively correlated with DAO, trypsin and lipase in the present study. *Romboutsia* has been studied to have a multifunctional metabolic capacity with carbohydrate utilization and single amino acid fermentation [40]. *L. chiayiensis AACE3* increased the levels of *Romboutsia*, enhancing the activity of intestinal digestive enzymes and thus improving feed utilization in chickens. Therefore, these findings suggest that *L. chiayiensis* AACE3 has the potential to replace aureomycin as a more environmentally friendly feed additive in poultry farming. However, some shortcomings remain in this study, such as some regional limitations of the study materials used and an incomplete study of the molecular mechanisms of growth promotion in *L. chiayiensis*. In the future, we will further investigate the way in which *L. chiayiensis*-mediated microbial changes regulate the alteration of host physiological phenotypes through metabolomics and transcriptomics. Overall, this study provides a new possibility for antibiotic replacement in poultry farming.

## 4. Materials and Methods

### 4.1. Bacterial Strains and Culture Conditions

*L. chiayiensis* AACE3 (Genbank No. CP107523.1) was isolated from fermented blueberry sap. It is now stored in the Faculty of Life Science and Technology, Kunming University of Science and Technology, Kunming City, China. For the individual colonies used for routine use, the strain was grown and passaged in lactic acid bacteria selective medium (MRS) broth at 37 °C for 24 h. Probiotics were harvested via centrifugation at 6,000 × g for 10 min, and after washing twice with phosphate-buffered saline (PBS, pH 7.2), the precipitate was resuspended in sterile PBS solution. One milliliter of the bacterial suspension solution was taken, the final concentration of probiotics was adjusted using the optical density method (spectrophotometer, LAMBDA 850, Waltham, MA, USA), and the exact numbers (1 × 10^9^ CFU/mL) were determined for later feed preparation.

### 4.2. Birds, Diet, and Experimental Design

Nandan-Yao chickens used in this experiment were purchased from Kunming Lihua Animal Husbandry Co., Ltd. The Animal Care and Use Committee of the Kunming University of Science and Technology approved the study protocol and provided the guidelines for this study. 

A total of 240 one-day-old Nandan-Yao chickens were randomly distributed into four treatment groups and fed for 21 days. Each group contained 6 replicates, with 10 chicks per replicate. The dietary groups included: (1) standard basal diet (NC group), (2) basal diet with antibiotic aureomycin (100 mg/kg of feed) (PC group), (3) basal diet with a low dose of *L. chiayiensis* AACE3 (1 × 10^6^ CFU/g of feed) (LC6 group), and (4) basal diet with a high dose of *L. chiayiensis* AACE3 (1 × 10^7^ CFU/g of feed) (LC7 group), 10^6^ CFU/g and 10^7^ CFU/g probiotic concentrations. It was confirmed by diluting and freeze-drying 10^9^ CFU/mL of probiotic solution and then taking 1 gram of lyophilized powder for smear plate testing. The lyophilized powder was then added to the basal feed at 0.1%. At 0, 7, 14, and 21 days, all birds and feed were weighed, and feed consumption and body weight were recorded weekly for each group. The body weight (BW), average daily feed intake (ADFI), average daily gain (ADG), and feed conversion ratio (FCR) were calculated accordingly. The basal diet was composed of soybean meal, corn, soybean oil, vitamins, and mineral premix and made per the (National Research Council (NRC) standard) [41] (Table 2). 

### 4.3. Sample Collection

On day 21, 6 chickens were randomly selected from each replicate and samples from the same replicate group were mixed for subsequent testing. Sampling of all chickens was performed on a sterile table in the chicken house and was performed on the same morning. All samples were placed in ice boxes during collection; the entire sample collection process took 3 hours and they were transferred to the laboratory within 20 min of completion for the next step. Blood samples were collected through the jugular vein, centrifuged at 3500× *g* and at 4 °C for 10 min to remove the supernatant, and then stored at −20 °C. The blood samples were collected through the jugular vein. The liver samples were collected into sterile tubes and stored at −20 °C. The duodenal digest was stripped and sonicated after tenfold dilution with PBS (pH 7.0), and the resulting digest sample was centrifuged at 15,000× *g* and 4 °C for 30 min. The supernatants were divided into aliquots and stored at −20 °C for enzymatic assays. From each bird, samples of the jejunum were collected, flushed with saline, and placed in a 4% formaldehyde solution for morphometric index evaluation. Cecum feces were collected in sterile plastic tubes, and samples from each chicken were combined and divided in the same proportions. The samples were stored at −20 °C until microbiota analysis.

### 4.4. Analysis of the Gut Morphostructure

The removed chicken jejunum was fixed in 4% paraformaldehyde and embedded in paraffin. Then, a ZEISS microscope (Shanghai, China) and ImageJ software (Version 1.8.0.345 Chinese, Kunming, China) were employed to perform histomorphometric examinations of 5 μm thick serial sections of the jejunal sample that were stained with hematoxylin and eosin (HE). Villus height (VH), crypt depth (CD), and VH:CD ratio measurements were obtained from at least 3 slides, with a minimum of 6 well-oriented indices to calculate their respective averages.

### 4.5. Measurement of Physiological Indicators

Precollected hepatic tissues were homogenized with 4 volumes of reagent kit extracts and centrifuged at 1000× *g* for 5 min. The supernatant was used for malondialdehyde (MDA, Suzhou Grace Biotechnology Co., Ltd., Suzhou, China) content, catalase (CAT, Suzhou Grace Biotechnology Co., Ltd., Suzhou, China) activity, superoxide dismutase (SOD, Suzhou Grace Biotechnology Co., Ltd., Suzhou, China) activity, total antioxidant capacity (T-AOC, Suzhou Grace Biotechnology Co., Ltd., Suzhou, China) activity, and glutathione reductase (GSH-Px, Suzhou Grace Biotechnology Co., Ltd., Suzhou, China) activity evaluation by using commercial kits. Stored sera were tested for immunoglobulin A (IgA), immunoglobulin G (IgG), interleukin-6 (IL-6), interleukin-1β (IL-1β) and tumor necrosis factor alpha (TNF-α) levels using the immune factor test kit (Quanzhou Jubang Biotechnology Co., Ltd., Quanzhou, China). Digestive fluid was collected from the duodenal intestine, and changes in amylase, trypsin and lipase levels were measured using an ELISA micro α-amylase activity assay kit, a trypsin activity assay kit and a lipase (LPS trace method) activity assay kit (Quanzhou Jubang Biotechnology Co., Ltd., Quanzhou, China).

### 4.6. Analyses of the Cecal Microbiota

From precollected cecum contents, a total of 12 samples were used for microbiome analysis. Microbial total genomic DNA was extracted from samples using the MagMAXTM-96 DNA multi-sample kit (Thermo Fisher Scientific, Suzhou New District, China) according to the manufacturer’s instructions. The 16S rRNA V3–V4 region was amplified and sequenced at the Majorbio Bio-Pharm Technology Co., Ltd (Shanghai, China) Genomic Analysis Platform-IBIS using Illumina MiSeq paired-end technology. Sequences were analyzed in the Ubuntu terminal using the UPARSE method [42], merging raw reads, filtering with the maximum expected error threshold at 1.0, dereplicating and mapping reads into operational taxonomic units (OTUs). Moreover, operational taxonomic units (OTUs) were clustered using Uparse based on the similarity threshold being greater than 97%. The RDP classifier algorithm was used to compare the 97% similar OTU representative sequences with the SILVA (Version 138, Shanghai, China) database for taxonomic analysis. The analyses related to the 16S data (i.e., diversity, phylum level, genus level, and correlation analysis) were performed on the Megabio online analysis platform (Shanghai, China).

### 4.7. Statistical Analysis

The mean ± standard error of the mean (SEM) values of all experimental data are determined from at least three independent replicates. GraphPad Prism 8.0 (GraphPad Software, Kunming, China) was used to conduct statistical analyses. The level of significance was set at * *p* < 0.05, ** *p* < 0.01, *** *p* < 0.001, and one-way ANOVA or non-differential/mixed comparisons were carried out to calculate statistical significance.

## 5. Conclusions

Reducing the overuse of antibiotics in poultry farming is one of the keys to curbing the spread of antibiotic resistance. Probiotics are a potential alternative to antibiotics in animal feed. The results of this study reveal the great potential of *L. chiayiensis* AACE3 as an antibiotic alternative in broiler farming. Supplementation with *L. chiayiensis* AACE3 enhances the health of chicks by increasing antioxidant capacity and modulating the levels of immune factors in the chicks. *L. chiayiensis* AACE3 also alters the gut microbial community of chicks, increasing the abundance of several bacterial species associated with inflammation, immunity and growth, such as *Akkermansia*, *Romboutsia*, and *Subdoligranulum*, among other beneficial microorganisms. Thus *L. chiayiensis* AACE3 can promote a host-friendly gut environment by changing the composition of gut microorganisms to improve health. Altogether, these findings suggest that *L. chiayiensis* AACE3 may be a suitable alternative to antibiotics in broiler production.

## Figures and Tables

**Figure 1 antibiotics-12-01356-f001:**
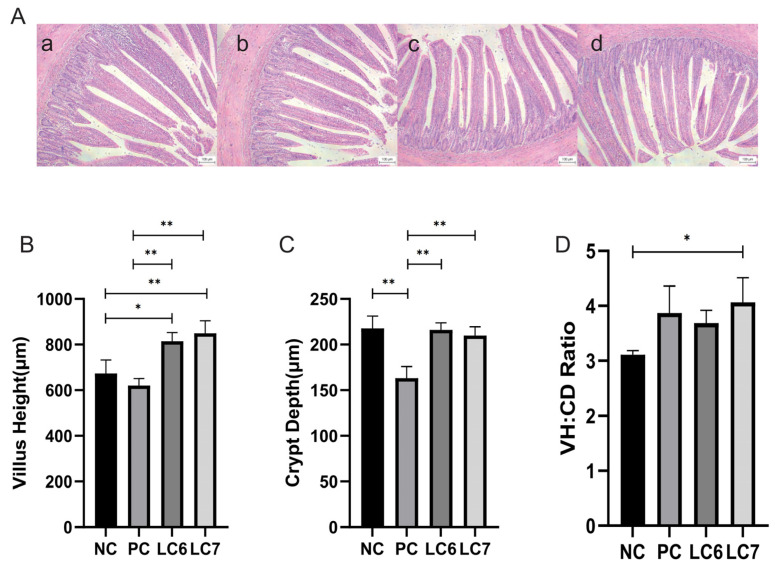
Effect of *L. chiayiensis* AACE3 group on jejunum morphology of chickens. (**A**) Jejunal histological samples by the groups (**a**)**:** NC, (**b**)**:** PC, (**c**): LC6, (**d**): LC7. (**B**) villus length, (**C**) crypt depth, (**D**) VH:CD ratio. Scale bars = 100 μm. * *p* < 0.05, ** *p* < 0.01.

**Figure 2 antibiotics-12-01356-f002:**
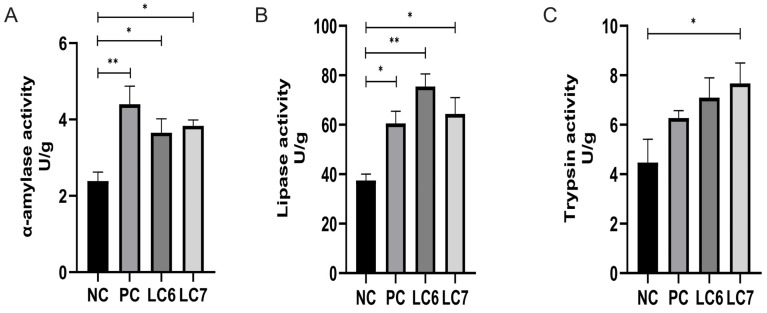
Effect of supplementation with *L. chiayiensis* AACE3 on duodenal digestive enzymes. (**A**) α-amylase activity. (**B**) Lipase activity. (**C**) Trypsin activity. * *p* < 0.05, ** *p* < 0.01.

**Figure 3 antibiotics-12-01356-f003:**
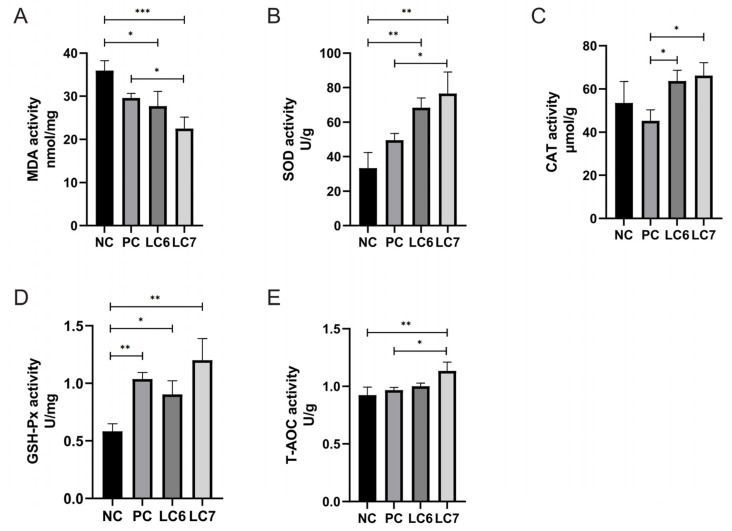
Influence of *L. chiayiensis* AACE3 on antioxidant enzyme activity and MDA levels in chickens. (**A**) MDA level content. (**B**–**E**) level contents of SOD, CAT, GSH-Px, and T-AOC. * *p* < 0.05, ** *p* < 0.01, *** *p* < 0.001.

**Figure 4 antibiotics-12-01356-f004:**
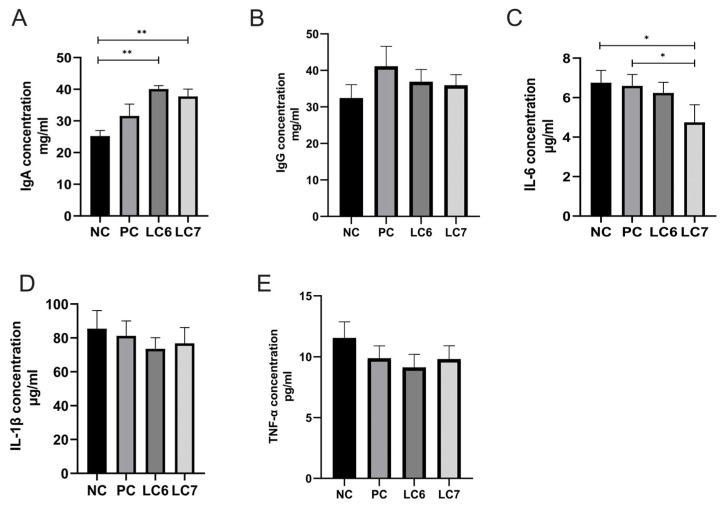
Induction of serum immune factors by *L. chiayiensis* AACE 3. (**A**,**B**) are immunoglobulin IgA and IgG levels. (**C**–**E**) are the levels of inflammatory factors IL-6, IL-1β, and TNF-α. * *p* < 0.05, ** *p* < 0.01.

**Figure 5 antibiotics-12-01356-f005:**
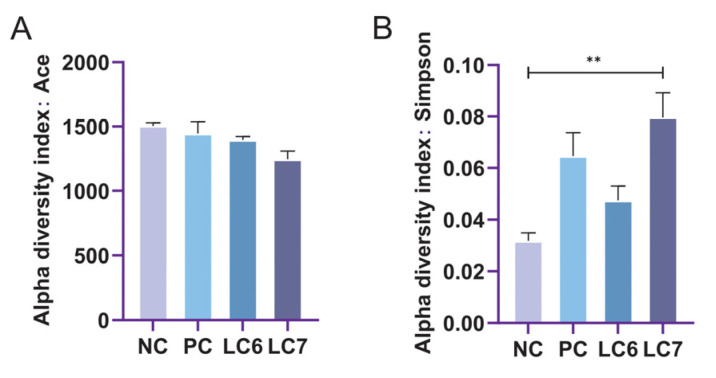
Alpha diversity of gut microbiota. (**A**) Ace. (**B**) Simpson. ** *p* < 0.01.

**Figure 6 antibiotics-12-01356-f006:**
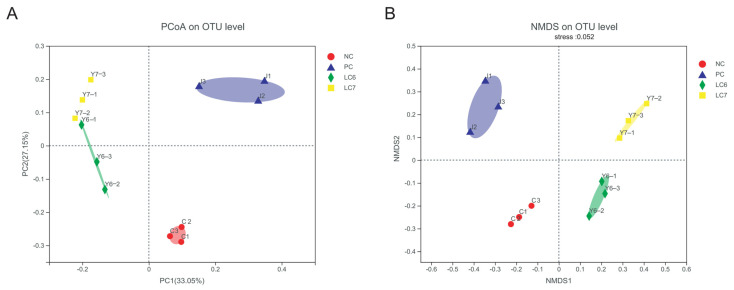
Beta diversity of gut microbiota. (**A**) PCoA. (**B**) NMDS.

**Figure 7 antibiotics-12-01356-f007:**
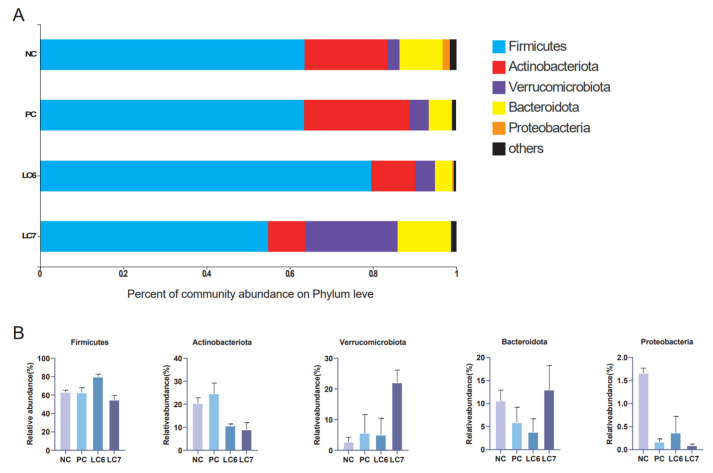
Composition of the phylum of the gut microbiota in chickens. (**A**) Relative abundance of the first five phyla. (**B**) Comparative differences at the phylum level of the gut microbiota.

**Figure 8 antibiotics-12-01356-f008:**
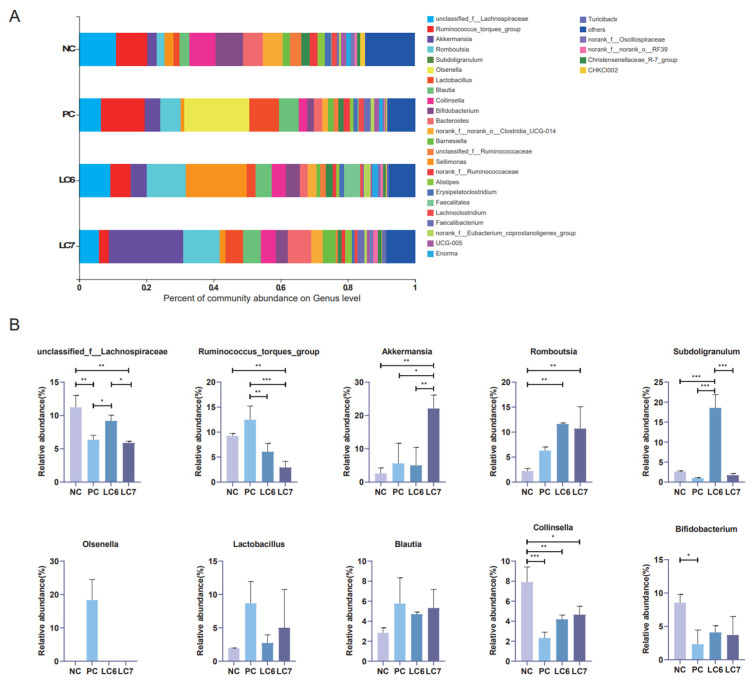
Composition of the genus of the gut microbiota in chickens. (**A**) Relative abundance of the first 30 genera. (**B**) Comparative genus-level differences in the top 10 gut microbiota. * *p* < 0.05, ** *p* < 0.01, *** *p* < 0.001.

**Figure 9 antibiotics-12-01356-f009:**
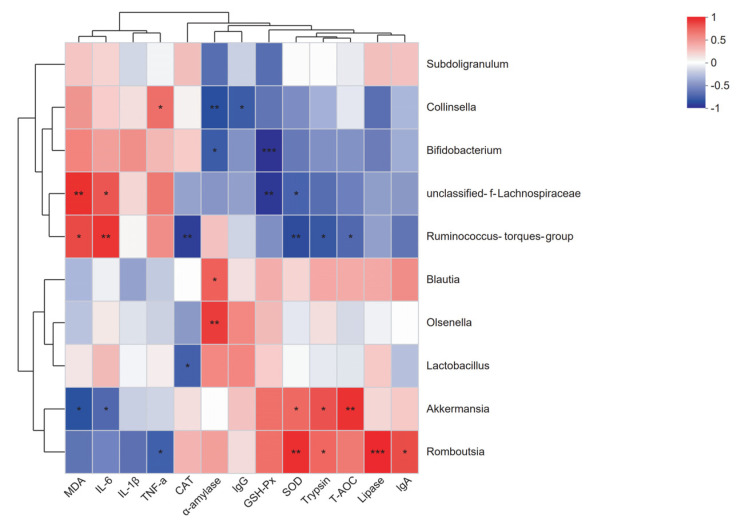
Heat map of correlation coefficients between gut microbiota and digestive characteristics, antioxidant enzyme activity and immune factors of chickens at 21 days.” red” indicates a positive correlation (*p* < 0.05) and “blue” indicates a negative correlation (*p* < 0.05). * *p* < 0.05, ** *p* < 0.01, *** *p* < 0.001.

**Table 1 antibiotics-12-01356-t001:** Effect of different doses of *L. chiayiensis* AACE 3 on growth performance of Nandan-Yao chickens from d 1 to 21 of the experiment.

Variables	NC	PC	LC6	LC7	SEM
BW (g)	135.12 ± 4.86 ^a^	146.40 ± 10.32 ^a^	139.68 ± 3.14 ^a^	149.81 ± 6.37 ^a^	3.302
ADFI (g)					
d 1 to d 21	12.19 ± 0.16 ^a^	13.59 ± 0.60 ^b^	12.81 ± 0.27 ^ab^	13.57 ± 0.47 ^ab^	0.719
ADG (g)					
d 1 to d 21	4.72 ± 0.07 ^a^	5.67 ± 0.09 ^b^	5.08 ± 0.15 ^ac^	5.61 ± 0.12 ^b^	0.226
FCR (g:g)					
d 1 to d 21	2.58 ± 0.03 ^a^	2.40 ± 0.05 ^a^	2.52 ± 0.06 ^a^	2.41 ± 0.04 ^a^	0.044

The values are given as mean ± SEM. ^a,b,c,^ means in each raw with different superscripts are significantly different, means not significantly different with the same letter (*p* < 0.05). SEM: pooled standard error of the mean. BW: body weight; ADFI: average daily feed intake; ADG: average daily gain; FCR: feed conversion ratio between the NC group and all experimental groups.

**Table 2 antibiotics-12-01356-t002:** Ingredient composition and nutrient contents of basal diets.

Ingredients, %	1–21 d
Corn	55.41
Soybean meal	31.50
Palm oil	5.00
Phosphorus	3.60
Calcium	1.30
Salt	0.34
Lysine HCL	1.40
Methionine	0.22
Arginine	0.03
Vitamin–mineral premix	0.50
Limestone	0.60
Sodium carbonate	0.10
Metabolizable energy (MJ·kg^−1^)	14.01
Crude protein	20.0
Calcium	1.00
Total phosphorus	0.55
Lysine total	1.41
Methionine	0.50

Supplied per kilogram of diet: vitamin A (trans-retinyl acetate), 10,050 IU; vitamin D3, 2800 IU; vitamin E (DL-α-tocopheryl acetate), 50 mg; vitamin K3, 3.5 mg; thiamine, 2.5 mg; riboflavin, 7.5 mg; pantothenic acid, 15.3 mg; pyridoxine, 4.3 mg; vitamin B12 (cyanocobalamin), 0.02 mg; niacin, 35 mg; choline chloride, 1000 mg; biotin, 0.20 mg; folic acid, 1.2 mg; Mn, 100 mg; Fe, 85 mg; Zn, 60 mg; Cu, 9.6 mg; I, 0.30 mg; Co, 0.20 mg; and Se, 0.20 mg.

## Data Availability

The datasets presented in this study can be found in online repositories. The names of the repository/repositories and accession number(s) can be found at: https://www.ncbi.nlm.nih.gov/ (accessed on 1 June 2023), CP107523.1, SRR23693698, SRR23693697, SRR23693694, SRR23693693, SRR23693693, SRR23693691, SRR23693690, SRR23693689, SRR23693687, SRR23693688, SRR23693696, and SRR23693695.

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
