# Peer review of "Effects of Dietary Supplementation of Lacticaseibacillus Chiayiensis AACE3 on Hepatic Antioxidant Capacity, Immune Factors and Gut Microbiology in Nandan Yao Chicks"

_antibiotics, 2023, doi:10.3390/antibiotics12091356_

Round 1

Reviewer 1 Report

The authors provide evidence that supplementation of Nandan Yao chickens with Lacticaseibacillus chiayiensis AACE3 supports the reprogramming of their intestinal microbiota and thus improves their anti-oxidative capacities, immunity, and performance. It is well established that the composition of gut microbiota is closely related to inflammatory cytokine secretion, which is surprisingly reminiscent of the inflammatory comorbidities and cytokine storm caused by viral or bacterial infections. In a carefully done comparative study the authors showed that the quality of the intestinal microbiota, the intestinal morphology, the liver antioxidative capability, and the content of relevant immune factors benefitted when the chickens were treated with probiotics. On this basis, the study adds another piece of evidence that the dietary supplementation of probiotics in poultry production can minimize or even replace therapies with antibiotics, such as with the tetracyclin aureomycin.

The experimental work has been done carefully with appropriate methodologies and statistical means. I understand well that the authors are enthusiastic about their results and their approach, but some statements could be downscaled without losing the main story and outcome. The authors should recheck these over-positive statements, such as the first sentence in the abstract, to name one example. Otherwise, the manuscript is fine and certainly merits publication.

Reviewer 2 Report

Publication in an international journal requires that the following conditions are met - that the work has some novelty, scientifically sound and presents new information or interpretation in such a way to be of interest to a wider audience than just those working in a region. This research meets the first two requirements, but not the third one. In this study, a local (very) slow growing birds was used – which reaches a body weight of just ~135-140 g at day 21. By comparison, modern broiler strains reach this body weight within day 7. Though the test probiotic shows beneficial effects, international relevance of a study using such a slow-growing bird is questionable. This is clearly of regional relevance. In future studies, make sure that the interest of the work for other types of chickens is made obvious

Other issues include,

L106: how was the dose levels (to be tested) determined?. How was the probiotic added??

L113: why NRC recommendations were used for such a slow-growing bird?. Thus the NC diet is not nutritionally negative.

L114-9: redundant. Delete.

Table 1: Check feed composition tables in international papers and improve. Consult a poultry nutritionist. Column 1: which amino acid? Stone powder?? 1% vitamins – not correct!!!. Why a range for Ca, NaCl and met????

L126: then only 3 replicates – inadequate (see also L181). This is another flaw in the stdy

L152: location needed for companies – throughout.

Table 2: Pooled SEM must be provided, not SD+/- treatment mean.

All figures: SD or SE ???

Lot of work has been done in this study. But there are flaws and I am reluctant the acceptance in the present form - for the reasons listed above.

Quality of language is fine

Round 2

Reviewer 2 Report

See my report. Major changes needed. I do not need to see the revised version.

Some revision and improvement has been done – but I am disappointed that the referee report was not respected and several issues were ignored.

The paper still has excessive number of language, spelling and presentation errors. I will leave these to the authors and the  Editorial Office to address.

L31-2: Not clear - delete

ADD ‘Nandon Yao’  in the title and abstract.

L77: ‘The good body condition of Nandan Yao chickens makes them a good material for studying chick health [20]. Very poor citation. I still believe that this slow-growing bird is a poor material for any study. The applicability of data to contemporary strains is poor. Please comment.

L114: why NRC recommendations were used for this very slow-growing bird?. Thus the NC diet is not nutritionally negative. I don’t see any explanation forthis..

L181: mean +/- SD is not SEM. Revise.

Table 2: Pooled SEM must be provided, not SD+/- treatment mean.

For all figures: SD or SE ??? – be specific.

L414: ..excessive obesity …??? In this strain?? Revise or delete.

A problem in the DISCUSSION is that it is wordy and redundant statements are included.

Round 3

Reviewer 2 Report

The paper could be accepted

Can be improved